# The Monte Carlo Method and New Device and Architectural Techniques for Accelerating It

## Abstract

Computing systems interacting with real-world processes must safely and reliably process uncertain data. The Monte Carlo method is a popular approach for computing with such uncertain values. This article introduces a framework for describing the Monte Carlo method and highlights two advances in the domain of physics-based non-uniform random variate generators (PPRVGs) to overcome common limitations of traditional Monte Carlo sampling. This article also highlights recent advances in architectural techniques that eliminate the need to use the Monte Carlo method by leveraging distributional microarchitectural state to natively compute on probability distributions. Unlike Monte Carlo methods, uncertainty-tracking processor architectures can be said to be *convergence-oblivious*.

## 1 Introduction

Uncertainty arises when systems carry out computations based on *measurements* (aleatoric uncertainty) or *limited knowledge* (epistemic uncertainty). Uncertainty introduces risk to actions taken based on measurements or limited knowledge. Studying and quantifying how uncertainty propagates through computations is a requirement when making principled decisions about the suitability of an uncertain system for an application.

Despite the importance of quantifying and understanding uncertainty, computer architectures and circuit implementations lack numerically-robust and computationally-efficient methods to programmatically process and reason about uncertainty. State-of-the-art techniques often employ the Monte Carlo method [1, 2, 3, 4] to estimate the effect of long sequences of arithmetic operations on inputs that are uncertain when closed-form propagation of uncertainty is not possible. Monte-Carlo-based methods can be sample-inefficient: the variance in the result of Monte Carlo integration using $n$ samples scales as $\frac{1}{\sqrt{n}}$ [3]. This means that if we wanted to halve the variance, we would need to *quadruple* the number of samples.

This article presents a framework for describing Monte-Carlo-based methods (Section 2). The framework poses them as the application of three steps: sampling, evaluation, and post-processing. In Section 3 we describe recent advances in physics-based programmable non-uniform random variate generators (PPRVGs) which can improve the sampling phase of Monte Carlo methods. Section 4 shows how a novel uncertainty-tracking microarchitecture, Laplace [5, 6], can provide a more efficient way to represent and compute on uncertain variables. Section 6 compares the performance of Laplace to the traditional Monte Carlo method.

Submitted to the Second Workshop on Machine Learning with New Compute Paradigms at NeurIPS (MLNCP 2024). Do not distribute.

## 2 The Monte Carlo Method

The phrase *Monte Carlo method* refers to a wide class of computational methods that sample from random variables to calculate solutions to computational problems. The earliest example of the use of a Monte Carlo method is attributed to Georges-Louis LeClerc [4], Comte de Buffon, who, in the eighteenth century, simulated a value of $\pi$ by dropping needles onto a lined background. He showed that when the needle has the same length as the distance between parallel lines, the probability that a randomly-thrown needle will overlap with a line is $\frac{2}{\pi}$. Therefore, $\pi$ can be estimated by throwing a large number of needles and averaging the number of times they overlap with a line.

### 2.1 The Monte Carlo Method: Sampling, Evaluation, and Post-Processing

The Monte Carlo method approximates a desired numerical property of the outcome of transformations of random variables. Practitioners use the Monte Carlo method when the desired property is not available analytically or because the analytical solution is computationally expensive. The desired property could be the expectation of the resulting random variable (*Monte Carlo integration*), a sample from it (*Monte Carlo sampling*), or its probability density function (*Monte Carlo simulation*).

Suppose that we want to obtain a property from the random variable $Y$ that is defined by transforming the random variable $X$ using the transformation $f : X \to Y$ (i.e., $Y = f(X)$). We summarize the steps of the Monte Carlo method to approximate the desired numerical properties as follows:

1. **Sampling:** The Monte Carlo method first generates i.i.d. samples from $X$. Let $n$ denote the number of samples of the random variable $X$ in the set $\{x_i\}_{i=1}^{n}$. This step typically uses a random number generator program running on a computer that can generate pseudo-random numbers from a uniform distribution. Samples from more complex random variables are generated using *Monte Carlo sampling*, where the Monte Carlo method itself is used to generate samples by transforming the uniform random variates. Examples of Monte Carlo sampling include the Box-Muller method [7] for generating standard Gaussian samples, inverse transform sampling for sampling from random variables for which an inverse cumulative distribution function (ICDF) exists[1], and Markov Chain Monte Carlo (MCMC) for more complex random variables [2].

   As an alternative to Monte Carlo sampling, we can use physical hardware to efficiently sample from a non-uniform random variable. Section 3 presents several such methods from the research literature which can provide large-batch single-shot convergence-oblivious random variate generation by exploiting physical processes that generate *non-uniform* entropy and can be sampled in parallel.

2. **Evaluation:** The second step of the Monte Carlo method then evaluates the transformation $f$ on the set of samples $\{x_i\}_{i=1}^{n}$ to obtain a set $\{y_i\}_{i=1}^{n}$ of $n$ samples of $Y$, where each $y_i = f(x_i)$. This step is called *Monte Carlo evaluation*.

   In the Monte Carlo method, the evaluation step is carried out on each sample $x_i$, one at a time. Section 4 presents recent research on computer architectures that can process compact representations of entire distributions at once, rather than one sample at a time as is the case for the traditional Monte Carlo method.

3. **Post-processing:** In the third and final step, the Monte Carlo method approximates the desired numerical property from the samples $\{y_i\}_{i=1}^{n}$ by applying an operation on their set. For example, taking their average (as in the case of Monte Carlo integration), applying the identity function (as in Monte Carlo sampling), or generating a representation of the probability density function, such as a histogram (as in Monte Carlo simulation).

## 3 Physics-Based Programmable Non-Uniform Random Variate Generation

Section 2 described the sampling of (possibly non-uniform) random variables as the first step of the Monte Carlo method. Most computing systems use pseudo-random number generators to generate uniform random variates. Computers generate samples from non-uniform random variables by using Monte Carlo sampling (Section 2). Since Monte Carlo methods could require large numbers of samples, these methods can be computationally-expensive and can lead to a significant overhead.

---

[1]Leemis *et al* [8] provides a good source of relationships between univariate random variables.

Two recent methods of generating non-uniform random variates from physical processes, Spot [9] and Grappa [10] have the following key features:

- They can *efficiently* generate *non-uniform* random variates: Spot, for example, can generate Gaussian random variables $260\times$ faster than the Box-Muller transformation running on an ARM Cortex-M0+ microcontroller, while dissipating less power than such a microcontroller.

- They are *physics-based*: Spot generates random variates using electron tunneling noise, while Grappa exploits the transfer characteristics of Graphene field-effect transistors.

- They are programmable: The distributions from which they can sample from are not fixed; their host systems can dynamically and digitally configure them to produce samples from a required probability distribution.

Due to these features, we call methods such as Spot and Grappa physics-based programmable non-uniform random variate generators (PPRVGs).

**Spot:** Spot is a method for generating random numbers by sampling a one-dimensional distribution associated with a Gaussian voltage noise source [11]. Using an analog-to-digital converter (ADC), Spot takes measurements of a physical process that generates Gaussian noise. Spot then maps this physically-generated univariate Gaussian to any other univariate Gaussian using only two operations: a multiplication and an addition [9]. Samples from any other non-uniform random variable are generated by creating a mixture of Gaussians.

**Grappa:** Grappa is a Graphene Field-Effect Transistor (GFET)-based programmable analog function approximation architecture [12]. Grappa relies on the non-linear transfer characteristics of GFETs to transform a uniform random sample into a non-uniform random sample [12].

Grappa implements a linear least-squares Galerkin approximation [13] to approximate the ICDF of a target distribution and carry out inverse transform sampling. The required orthonormal basis functions are obtained from the GFET transfer characteristics using the Gram-Schmidt process [14].

Tye *et al.* showed that Monte Carlo integration using samples generated by Grappa is at least $1.26\times$ faster than using a C++ lognormal random number generator. Subsequent work [12] demonstrated an average speedup of up to 2x compared to MATLAB for lognormal, exponential, generalized Pareto, and Gaussian mixture distributions, with the execution time independent of the target distribution.

## 4    Beyond the Monte Carlo Method

Let $X$ be a random variable and $f : X \to Y$ be a transformation of $X$. Denoting the resulting random variable as $Y = f(X)$, from the change of variable formula for random variables (Theorem 1 in Appendix A), we obtain probability density function $p_Y$ of $Y$. If $p_X$ is the probability density function of $X$, then the probability density function of $p_Y$ of $Y$ is,

$$p_Y(y) = p_X \circ f^{-1}(y) |\det \nabla f^{-1}(y)|, \tag{1}$$

where $y \in Y$, and $\nabla_y f^{-1}$ is the Jacobian matrix. Using the change of variables technique of integrals, we obtain

$$
\begin{aligned}
\mathbb{E}_{p_X}[f(X)] &= \int_X f(x) p_X(x) \, \mathrm{d}x && \text{by Equation 7 in Appendix A} \\
&= \int_Y y p_Y \circ f^{-1}(y) |\det \nabla f^{-1}(y)| \, \mathrm{d}y && \text{by change of variables (integration)} \\
&= \int_Y y p_Y(y) \, \mathrm{d}y && \text{by Theorem 1 in Appendix A} \\
&= \mathbb{E}_{p_Y}[Y].
\end{aligned}
$$

Thus, if we had access to $p_Y$ of $Y$, we can evaluate $\mathbb{E}_{p_X}[f(X)]$ by taking the expectation of the random variable $Y$ with respect to the $p_Y$. When $p_Y$ isn't directly accessible, we usually obtain the expectation of $Y$ by using Monte Carlo integration. However, having access to $p_Y$ would eliminate the need to use the Monte Carlo method completely.

Laplace [5, 6] is a computer microarchitecture that is capable of directly computing $p_Y$ by representing distributional information in its microarchitectural state and tracking how these distributions evolve under arithmetic operations, transparently to the applications running on it. Laplace provides a representation for the distribution (see Definition 3 in Appendix A) of random variables, and carries out *deterministic computations on this distribution*.

Laplace's in-processor distribution representation has an associated *representation size* that describes the *precision* at which the probability distribution is represented. Higher values of the representation size result in a more accurate representation. A useful analogy is the IEEE-754 standard for representing the uncountable infinite set of real numbers as floating-point numbers [15, 16] on a finite-precision computer.

Computer architectures such as Laplace eliminate the need for using the Monte Carlo method and can therefore have far-reaching consequences in areas where the Monte Carlo method is used. For example, to approximate the predictive Gaussian Process posterior distribution with an uncertain input, Deisenroth *et al* [17, 18] used moment-matching; Laplace could compute the posterior exactly, up to the precision of the representation.

## 5  Methods

The remaining text compares and evaluates *Monte Carlo methods* and *Laplace-based methods*. Both methods were evaluated on single-threaded applications written in the C programming language.

**Monte Carlo method:**  We use the standard Monte Carlo method that we described in Section 2. We use the pseudo-random number generator `rand` from the Standard C Library [19] to sample from uniform distributions and use the modified Box-Muller method [20] as implemented by the `gsl_ran_gaussian_ziggurat` function in the GNU Scientific Library [21]. We compile our code using `clang`, the C family front-end to LLVM [22] , with optimization set to `-O3`[2].

**Laplace:**  We use Laplace as a replacement for the Monte Carlo method, as described in Section 4. In our experiments, we exclusively use Laplace's Telescopic Torques Representation (TTR) [5] as provided by a commercial implementation of Laplace [23], release 2.6.

We compare these methods by empirically measuring and reporting the average *run time* and the average *Wasserstein distance [24] of the output to a ground truth* in two different applications of Monte Carlo simulation. We change the number of samples (for Monte-Carlo-based methods), or the representation size (for Laplace-based methods) to observe the trade-offs between accuracy and run time. For each configuration of number of samples or representation size, we repeat the experiments 30 times to account for variation in the process of sampling[3]. See Appendix C for more detail on our methods. Figure 1 summarizes our results.

### 5.1  Applications

We carry out the experiments described above on two applications of Monte Carlo simulation.

**Monte Carlo Convergence Challenge Example:**  Let $X^{\mathrm{con}}$ be the initial random variable that we sample from, with its probability density function $p_{X^{\mathrm{con}}}$ being a Gaussian mixture. given by:

$$p_{X^{\mathrm{con}}}(x) = 0.6 \left( \frac{1}{0.5\sqrt{2\pi}} e^{-2(x-2)^2} \right) + 0.4 \left( \frac{1}{1.0\sqrt{2\pi}} e^{\frac{-(x+1)^2}{2}} \right). \tag{2}$$

For the Monte Carlo evaluation step of Section 2, we define a function $f^{\mathrm{con}}$ as a sigmoidal function:

$$f^{\mathrm{con}}(x) = \frac{1}{1 + e^{-(x-1)}}. \tag{3}$$

---

[2]We will make all code necessary for exact replication of our experiments available through Github.

[3]We calculate the Wasserstein distance for Laplace's representation of the output distributions by generating 1,000,000 samples from the representation. Therefore, we also repeat the Laplace experiments 30 times for each representation size even though Laplace's uncertainty-tracking methods are deterministic and convergence-oblivious. The variation we see in the results in Figure 1 is therefore *only* due to sampling variance.

For the Traditional Monte Carlo method, we evaluated on $n \in \{4, 256, 1152, 2048, 4096, 8192, 16000, 32000, 128000, 256000\}$. For Laplace, we evaluated on $r \in \{16, 32, 64, 256, 2048\}$.

**Poiseuille's Law for Blood Transfusion**   As a real-world application, we use Poiseuille's Law, a mathematical model from fluid dynamics used to calculate the rate of laminar flow, $Q$, of a viscous fluid through a pipe of constant radius [25, 26]. This model is used in medicine as a simple method for approximating the rate of flow of fluids, such as blood, during transfusion [27, 28]. We look at Poiseuille's Law applied to the case of blood transfusion using a pump with the following parameters:

$\Delta P$: Pressure difference created by the pump, where $\Delta P \sim \mathcal{N}(5500000 \, \mathrm{mPa}, 36000^2)$.

$\mu$: Viscosity of the fluid, where $\mu \sim \mathcal{U}(3.88 \, \mathrm{mPas}, 4.12 \, \mathrm{mPas})$.

$l$: Length of the tube from the cannula to the pump, where $l \sim \mathcal{U}(6.95 \, \mathrm{cm}, 7.05 \, \mathrm{cm})$.

$r$: Radius of the cannula, where $r \sim \mathcal{U}(0.0845 \, \mathrm{cm}, 0.0855 \, \mathrm{cm})$.

We assume the cannula to have a gauge of 14 (a radius of $0.85 \, \mathrm{mm}$) and the viscosity of blood to be $4 \, \mathrm{mPas}$ [28]. [29] reported that for porcine blood, the uncertainty of using a ventricular assist device to measure blood viscosity in real time was $\pm 0.12 \, \mathrm{mPas}$; we use this as the uncertainty of the viscosity.

The flow rate $Q$ is therefore measured in $\mathrm{cm}^3/\mathrm{s}$. Using these parameters, we can calculate the flow rate using Poiseuille's Law:

$$Q = \frac{\pi r^4 \Delta P}{8\mu l}. \tag{4}$$

For the Traditional Monte Carlo method, we evaluated on $n \in \{4, 256, 1152, 4096, 8192, 32000, 128000, 256000, 512000, 640000\}$. For Laplace, we evaluated on $r \in \{16, 32, 64, 128, 256, 2048\}$.

# 6   Results

Figure 1 shows Pareto plots of the mean run time against the Wasserstein distance from the ground truth for both applications. A key observation is that the variance of the Laplace-based methods is more or less constant as we increase the representation size. Laplace carries out *deterministic computations on probability distributions*; this variance is caused by using a finite number of samples from Laplace's representation of the output distribution to calculate the Wasserstein distance. It is possible to calculate the Wasserstein distance directly from the Laplace processor representation but we did not do so at the time of writing. This calculation would be deterministic since it only depends on the representation of the distribution. In contrast, each run of the Monte Carlo method results in a different output distribution; to reduce this variance we need to increase the number of samples. In this way, Laplace is *convergence-oblivious* to the number of samples.

Increasing the representation size larger than $r = 32$ provides a worse trade-off with the run time for both applications. Table 1 shows that for the accuracy obtained by Laplace, the equivalent Monte Carlo simulation is $113.85\times$ (for the Monte Carlo Convergence Challenge example) and $51.53\times$ (for the Poiseuille's Law for Blood Transfusion application) slower. If much better accuracy is required, then the Monte Carlo method will need to be used. However, if the accuracy provided by Laplace is sufficient, it provides a potentially-orders-of-magnitude-faster alternative that is also *consistent outputs across repetitions*.

Tables with the numerical results are in Appendix F. Appendix F also compares histograms of the resulting distributions and provides additional discussion.

# 7   Conclusions

The Monte Carlo method is a powerful and historically-significant tool for solving complex problems that might otherwise be intractable. It involves three simple steps: sampling, evaluating and post-processing. Despite its versatility, the Monte Carlo method can suffer from inefficiencies. One of these is that generating samples for the first step of the Monte Carlo method is inefficient when samples are required from non-uniform probability distributions. Recent advances in physics-based random number generators, namely Spot [9, 11] and Grappa [12] address these challenges.

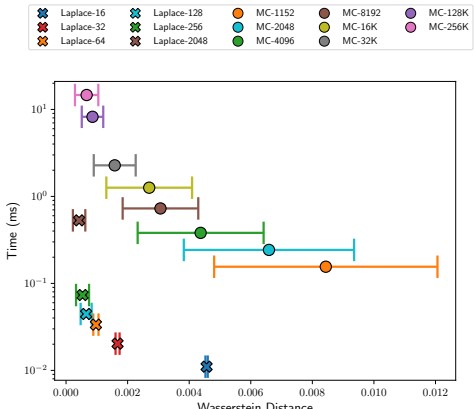
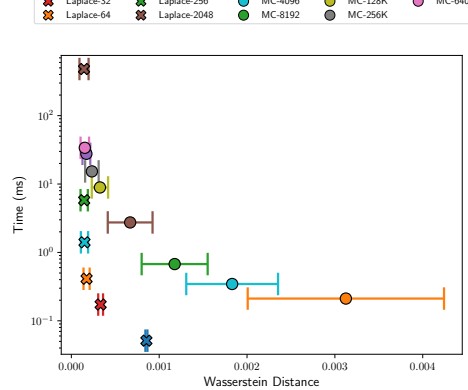

(a) Pareto plot for the Monte Carlo Convergence Challenge application from Section 5.1. We omitted $n = 4, 256$ for clarity; see Table 2 in Appendix F.

(b) Pareto plot for the Poiseuille's Law for Blood Transfusion application from Section 5.1. We have omitted $n = 4, 256$ for clarity.; see Table 3 in Appendix F.

Figure 1: Pareto plots between the mean run time, and the mean Wasserstein distance from the ground truth output distribution. The error bars show $\pm 1$ standard deviation. For the Monte Carlo Convergence Challenge example (a), Traditional Monte Carlo obtains similar accuracy to Laplace with $r = 32$ at 32,000 samples. For the Poiseuille's Law for Blood Transfusion application (b), Traditional Monte Carlo obtains similar accuracy than Laplace with $r = 32$ at 128,000 samples. In the legends, MC stands for *Traditional Monte Carlo, implemented in C*. We use the log scale on the vertical axis.

| Problem | Core | Representation Size / Number of samples | Wasserstein Distance (mean ± std. dev.) | Run time (ms) (mean ± std. dev.) |
|---|---|---|---|---|
| Monte Carlo Convergence Challenge | Laplace | 32 | $0.00167 \pm 0.00007$ | $0.020 \pm 0.004$ |
| Monte Carlo Convergence Challenge | Traditional Monte Carlo | 32000 | $0.00158 \pm 0.00068$ | $2.277 \pm 0.346$ |
| Poiseuille's Law for Blood Transfusion | Laplace | 32 | $0.00033 \pm 0.00003$ | $0.173 \pm 0.006$ |
| Poiseuille's Law for Blood Transfusion | Traditional Monte Carlo | 128000 | $0.00033 \pm 0.00009$ | $8.914 \pm 1.566$ |

Table 1: Results show the mean Wasserstein distance and the run time required the best overall configuration for Laplace and the close-to-equivalent results Monte Carlo configurations. For the Monte Carlo Convergence Challenge example, Traditional Monte Carlo takes approximately $113.85\times$ longer than Laplace with $r = 32$. For the Poiseuille's Law for Blood Transfusion application, Traditional Monte Carlo takes approximately $51.53\times$ longer than Laplace with $r = 32$.

Techniques such as Laplace [5, 6] represent probability distributions in a computing system using an approximate fixed-bit-width representation in a manner analogous to how traditional computer architectures approximately represent real-valued numbers using fixed-bit-width representations such as the IEEE-754 floating-point [15, 16] representation. The computations of Laplace are approximations of explicit Monte Carlo methods in much the same way that computations on floating-point are approximations of arithmetic on real numbers. Laplace does not require iterative and repeated processing of samples until convergence to a target distribution is achieved, nor does it suffer from the high variance observed across Monte Carlo runs. Methods like Laplace are therefore *convergence-oblivious*.

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

## Supplementary Material

## A Mathematical Preliminaries

To establish a consistent framework for discussing the Monte Carlo method, we first introduce key definitions and theorems that we will use throughout this article. Let $\mathbb{R}^+$ be the set of positive real numbers (i.e., $[0, \infty)$).

**Definition 1 (Probability density function)** *Let $X$ be a set and $p_X$ be a map from $X$ to $\mathbb{R}^+$,*

$$p_X : X \to \mathbb{R}^+,$$

*that satisfies:*

$$\int_X p_X(x) \, \mathrm{d}x = 1.$$

*We define $p_X$ to be the* probability density function *on $X$.*

**Definition 2 (Random Variable)** *Let $X$ be a set and $p_X$ a probability density function on $X$. We define the tuple $(X, p_X)$ as a random variable[4].*

Often, the set $X$ will be a real space $\mathbb{R}^d$. A random variable is a variable that can take on different values from its defining set $X$. The probability density function $p_X$ is a function that calculates how likely $X$ is to take on a *particular value* $x \in X$. Generating an instance value from a random variable is called *sampling from the random variable*.

For brevity, we use the following notation: an uppercase letter, such as $X$, denotes a random variable and the matching lowercase letter $x$ denotes an instance value of the random variable $X$. We denote a set of $n$ independently and identically distributed (i.i.d.) samples or variates of a random variable $X$ as $\{x_i\}_{i=1}^n$, where $i$ indexes this set and each $x_i$ is an instance value of $X$.

Let $f : X \to Y$ denote a *transformation* from a random variable $X$ to a random variable $Y$ (i.e., $Y = f(X)$)[5]. We can also apply $f$ to an instance value $x$ of $X$ to obtain an instance value $y = f(x)$ of $Y$.

The probability distribution of a random variable is a set function $\mathbb{P}_X : \Omega_X \to \mathbb{R}^+$ that tells us about the probability of the random variable taking on the values inside the given set, where $\Omega_X$ is a set of subsets of $X$. $\Omega_X$ must technically be a $\sigma$-algebra, but for our purposes, it can be thought of as a set of sets each of which can be expressed as a countable union of disjoint sets that also belong to $\Omega_X$[6]. Since we are only considering random variables that contain probability density functions, we define the distribution of a random variable as:

**Definition 3 (Distribution of a Random Variable)** *Given a random variable $X$ with probability density function $p_X$, the distribution of $X$ is given by the set function*

$$\mathbb{P}_X : \Omega_X \to \mathbb{R}^+$$

$$\omega = \bigcup_i U_i \mapsto \mathbb{P}_X(\omega) = \sum_i \int_{U_i} p_X(x) \, \mathrm{d}x, \tag{5}$$

where $U_i$ is a collection of disjoint sets whose union equals the input set $\omega$.

If $f$ is invertible and once-differentiable, then Theorem 1 derives the probability density function of $Y$, denoted as $p_Y$ [31, Chapter 3.7].

**Theorem 1 (Change of variables)** *Given a random variable $X$ with a probability density function $p_X$ and an invertible and once-differentiable transformation $f : X \to Y$, the probability density*

---

[4]There can be measure theoretic random variables, such as the random variable that has the Cantor distribution, that do not admit probability density functions [30]. In this article, we do not consider such exotic random variables.

[5]$f$ transforms the set $X$ such that there exists a valid probability density function $p_Y$ over the set $Y$.

[6]The sets in a $\sigma$-algebra must also satisfy that countable unions and intersections of arbitrary sets also belong to $\Omega_X$, along with $X$ and $\emptyset$.

function $p_Y$ of the random variable $Y = f(X)$ is given by:

$$p_Y : Y \to \mathbb{R}^+,$$
$$y \mapsto p_Y(y) = p_X \circ f^{-1}(y) |\det \nabla f \left( f^{-1}(y) \right)|^{-1}$$
$$= p_X \circ f^{-1}(y) |\det \nabla f^{-1}(y)|,$$

where $f^{-1}$ is the inverse of $f$ and $\nabla f(\,\cdot\,)$ and $\nabla f^{-1}(\,\cdot\,)$ denote the Jacobian matrices of $f$ and $f^{-1}$ respectively.

A key statistic that is often computed of a random variable is its expectation.

**Definition 4 (Expectation of a random variable)** *Given a random variable $X$ with probability density function $p_X$, we define the expectation $\mathbb{E}_{p_X}[X]$ of $X$ as*

$$\mathbb{E}_{p_X}[X] = \int_X x p_X(x) \, \mathrm{d}x. \tag{6}$$

Expectations can be calculated of transformations of random variables as well.

**Definition 5 (Expectation of a transformation of a random variable)** *Given a random variable $X$ with probability density function $p_X$ and a transformation $f : X \to Y$ from $X$ to a random variable $Y$[6], we define the expectation $\mathbb{E}_{p_X}[f(X)]$ of the random variable $f(X)$ as*

$$\mathbb{E}_{p_X}[f(X)] = \int_X f(x) p_X(x) \, \mathrm{d}x. \tag{7}$$

This is called the Law of the Unconscious Statistician [32].

# B    Buffon's Needle

In this section, we describe Buffon's Needle in more detail. Let $X$ be the random variable that denotes the location of a thrown needle and $f$ be a transformation on $X$ defined as:

$$f : X \to \{0, 1\},$$
$$x \mapsto f(x) = \begin{cases} 1 & \text{if needle lands on a line} \\ 0 & \text{otherwise.} \end{cases}$$

$f$ therefore identifies whether a dropped needle lands on a line. The resulting random variable $f(X)$ is a Bernoulli random variable $\mathrm{Bern}(p)$, where the probability of success $p$ is the probability of a needle landing on line. Since $\mathbb{E}_{f(X) \sim \mathrm{Bern}(p)}[f(X)] = p$, the expectation of $f(X)$ is precisely the probability of a needle landing on a line. The expectation of the resulting random variable $f(X)$ is $\frac{2}{\pi}$, as shown by LeClerc [4]. One can approximate this expectation by sampling from $X$ by dropping needles, evaluating $f$ by checking whether each needle landed on a line, and taking the average of the resulting samples of $f(X)$.

# C    Methods: Additional detail

## C.1    Measuring the run time

We measure the run time as the sum of the time taken to generate samples incurred during the sampling step of the Monte Carlo method or the initializing step of Laplace, and the time taken for the evaluation step. For both methods, we measure time using the `gettimeofday` function from the Standard C library [19]. We measure the time from the start of the main entry point until the end of key computations. The reported times omit any time spent by the programs on saving and reporting the results. We took further measures to ensure that our results were meaningful; these are detailed in the supplementary material.

In order to explicitly quantify the *post-processing* step of the Monte Carlo method, we compute the mean and the variance of the samples obtained from the Monte-Carlo-based experiments. Such a step is not necessary with Laplace, because Laplace already provides a usable representation of the output distribution. We note that we are being generous to the Monte Carlo method, since the mean and the variance alone does not fully capture the shape of a non-Gaussian distribution. In contrast, Laplace captures the full distribution in its representation.

## C.2 Measuring the Wasserstein Distance

The Wasserstein distance [24] is a metric that measures the distance between probability distributions. We quantify the distance of the outputs to the ground truth using the Wasserstein distance between the output distribution calculated by each approach and the ground truth output distribution. We compute the ground truth output distribution by running the Monte Carlo method with 1,000,000 samples. In our experiments, we calculate the Wasserstein distance using the `scipy.stats.wasserstein_distance` function from the `SciPy` Python package [33].

## C.3 Experimental setup

Let $n$ be the number of samples used in the sampling step of a Monte Carlo simulation. We perform experiments with various values of $n$ on an Apple M1 Pro with 16GB LPDDR5 RAM, running macOS 13.5.1. This provides a baseline for the performance of the Monte Carlo method that can be expected in the real-world.

Similarly, for Laplace, we varied the representation size $r$. Since the Laplace cores generate in-processor representations of the output distribution, we take samples from this distribution to compute the Wasserstein distance. We take 1,000,000 samples, similar to the ground truth. We do not include the time taken for this sampling in the wall-clock time because this sampling is done solely to calculate the Wasserstein distance and is not part of a typical use case of Laplace.

# D    Ensuring Meaningful Timing Results

When running the experiments on the Monte Carlo method, each repetition of an experiment was run after a 5s delay. This delay ensures that we avoid buffer cache optimizations carried out by the operating system.

We also note that we did not exploit parallelization when running Traditional Monte Carlo since the available implementation of Laplace did not exploit parallelization either. We felt that this provided an apples-to-apples comparison.

# E    Applications: Additional Detail

## E.1 Monte Carlo Convergence Challenge Example

Here, we present a more complete description of the Monte Carlo Convergence Challenge example. For ease of reading, we repeat the key equations.

Let $X^{\mathrm{con}}$ be the initial random variable that we sample from, with its PDF $p_{X^{\mathrm{con}}}$ being a Gaussian mixture. The underlying set of $X^{\mathrm{con}}$ is $\mathbb{R}$, and $p_{X^{\mathrm{con}}}$ is given by:

$$
\begin{aligned}
p_{X^{\mathrm{con}}}(x) = 0.6 &\left( \frac{1}{0.5\sqrt{2\pi}} \exp\left(-2(x-2)^2\right) \right) \\
+ 0.4 &\left( \frac{1}{1.0\sqrt{2\pi}} \exp\left( \frac{-(x+1)^2}{2} \right) \right).
\end{aligned}
\tag{8}
$$

For the Monte Carlo evaluation step of Section 2, we define a function $f^{\mathrm{con}}$ as a sigmoidal function:

$$
\begin{aligned}
f^{\mathrm{con}} &: X \to (0,1), \\
x &\mapsto f^{\mathrm{con}}(x) = \frac{1}{1 + e^{-(x-1)}}.
\end{aligned}
\tag{9}
$$

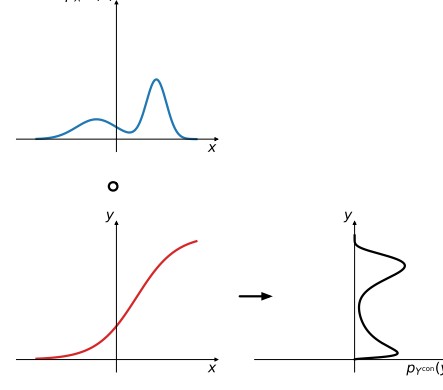

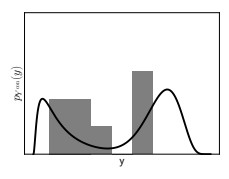

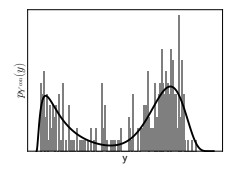

(b) Monte Carlo simula-
tion with 8 samples.

(c) Monte Carlo simula-
tion with 256 samples.

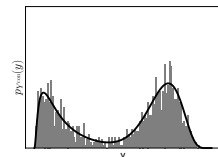

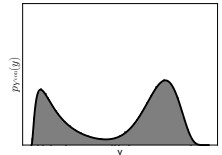

(a) Graphs of the PDF $p_{X^{\mathrm{con}}}$ (blue), the transfor-
mation $f^{\mathrm{con}}$ (red), and the output density function
$p_{Y^{\mathrm{con}}}$ (black) of the Monte Carlo Convergence Chal-
lenge application.

(d) Monte Carlo simula-
tion with 1024 samples.

(e) Monte Carlo simula-
tion with 128,000 sam-
ples.

Figure 2: Left: graphs of the analytical input and output distributions, and the evaluation function for
the Monte Carlo Convergence Challenge example. Right: histograms of the output of Monte Carlo
simulation with 8 (b), 256 (b), 1024 (c), and 128,000 (d) samples. The input PDF has two modes,
and the output distribution is heavily influenced by them (see black curve). If only a few samples
are used during Monte Carlo simulation, such as in (b), then the resulting histogram will be biased
toward a single mode.

Let $Y^{\mathrm{con}} = f^{\mathrm{con}}(X^{\mathrm{con}})$ denote the output random variable. The underlying set of $Y^{\mathrm{con}}$ is $(0, 1)$. Its
PDF $p_Y^{\mathrm{con}}(y)$ can be analytically calculated using Theorem 1:

$$
\begin{aligned}
p_{Y^{\mathrm{con}}}(y) = \Bigg( 0.6 &\left( \frac{1}{0.5\sqrt{2\pi}} \exp\left(-(\mathrm{logit}(y) - 2)^2\right) \right) \\
+ 0.4 &\left( \frac{1}{1.0\sqrt{2\pi}} \exp\left( \frac{-(\mathrm{logit}(y) + 1)^2}{2} \right) \right) \Bigg) \\
&\times \left| \frac{\exp(1 - \mathrm{logit}(y))}{(\exp(1 - \mathrm{logit}(y)) + 1)^2} \right|^{-1},
\end{aligned}
\tag{10}
$$

where $\mathrm{logit}(y)$ is:

$$
\mathrm{logit}(y) = 1 + \log \frac{y}{1 - y}.
\tag{11}
$$

. Figure 2 plots the functions of Equations 8, 9, and 10, and highlights the problem of uncertainty
propagation. The function $f^{\mathrm{con}}$ (red curve) transforms the random variable $X^{\mathrm{con}}$ (PDF shown as
the blue curve) into the random variable $Y^{\mathrm{con}}$ (PDF shown as the black curve). In particular, $f^{\mathrm{con}}$
transforms the two modes of $X^{\mathrm{con}}$ into the two modes of $Y^{\mathrm{con}}$.

We chose this application to showcase issues with convergence in traditional Monte Carlo simulation
(see Figure 2). Due to the multi-modal distribution $p_{X^{\mathrm{con}}}$, using too few samples could bias the
resulting histogram of the Monte Carlo simulation toward the largest mode and not represent the
other mode well, as in Figure 2b. The fidelity of the output of Monte Carlo methods is therefore
sensitive to the number of samples taken from $X^{\mathrm{con}}$, and to the shape of the function $f^{\mathrm{con}}$. By
contrast, uncertainty-tracking processors such as Laplace are not sample-based and can be said to be
*convergence-oblivious*.

# F   Additional Results and Discussion

We provide an abridged set of results and observations in this section. We have broken the discus-
sion into three sections for comparing the relationship between the Wasserstein distance and run
time, comparing the number of required dynamic instructions, and comparing histograms of output
distributions.

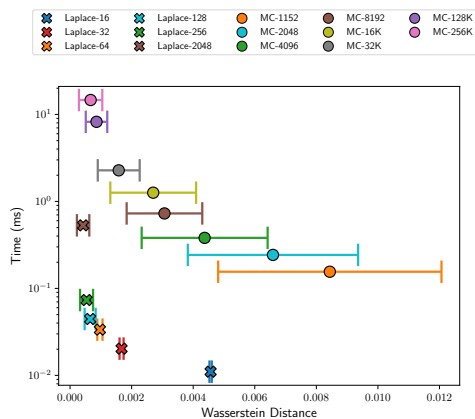 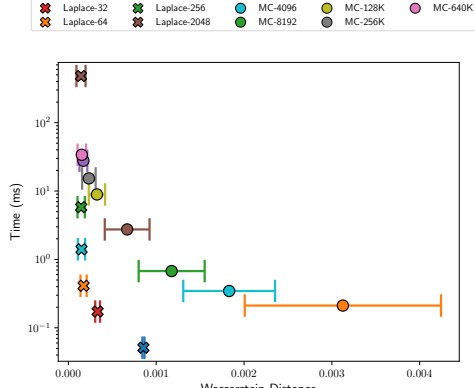

(a) Pareto plot for the Monte Carlo Convergence Challenge application from Section 5.1. We have omitted $n = 4,256$ for clarity.

(b) Pareto plot for the Poiseuille's Law for Blood Transfusion application from Section 5.1. We have omitted $n = 4,256$ for clarity.

Figure 3: Pareto plots between the mean run time, and the mean Wasserstein distance from the ground truth output distribution. The error bars show $\pm 1$ standard deviation, as in Tables 2-3. For the Monte Carlo Convergence Challenge application, (a) shows that Traditional Monte Carlo obtains better accuracy than Laplace with $r = 32$ (up to 1-standard deviation) after 128,000 samples. For the Poiseuille's Law for Blood Transfusion application, (b) shows that Traditional Monte Carlo obtains better accuracy than Laplace with $r = 32$ (up to 1-standard deviation) after 2048 samples. In the legends, MC stands for *Traditional Monte Carlo, implemented in C*. We use the log scale on the vertical axis.

## F.1 Comparing Wasserstein Distance and Run Time

Tables 2 and 3 show the means and the standard deviations of the run time and the Wasserstein distance[7] for the Monte Carlo Convergence Challenge and the Poiseuille's Law for Blood Transfusion examples, respectively. Figure 3 plots the run time and the Wasserstein distance across all experimental variations to show the Pareto boundary for each application. In general, these results match intuition, where an increase in $n$ improves the accuracy of the output distributions compared to the ground-truth distribution, at the expense of run time. We analyze the results for each application in turn.

### F.1.1 Monte Carlo Convergence Challenge

Table 2 and Figure 3a shows the key results for this application. A Laplace configuration with $r > 32$ improves the Wasserstein distance less than it worsens the run time. Therefore, we chose $r = 32$ to be the *best overall* configuration of Laplace for this application. We see the Monte Carlo method requires approximately 32,000 samples to obtain a similar Wasserstein distance to Laplace with $r = 32$. For this configuration, Laplace takes $113.85\times$ less time. To obtain an accuracy that is better than Laplace with $r = 32$ up to 1-standard deviation, the Monte Carlo method requires approximately 128,000 samples, for which it takes $411.25\times$ more time. Similarly, if we required the Monte Carlo method to obtain a Wasserstein distance better than Laplace with $r = 32$ up to 2-standard deviations, it would require approximately 256,000 samples, for which it takes $732.35\times$ more time.

### F.1.2 Poiseuille's Law for Blood Transfusion

Table 3 and Figure 3b show that the best trade-off between accuracy and run time is made by Laplace with $r = 32$. To match the mean accuracy of this configuration of Laplace, Traditional Monte Carlo requires $256,000$ samples. This takes $51.53\times$ more time than Laplace. To obtain an accuracy better than Laplace with $r = 32$ up to 1-standard deviation and 2-standard deviations, Traditional Monte Carlo requires approximately 512,000 samples. This takes $160.06\times$ more time than Laplace.

---

[7]The Wasserstein distances are of very different scales. The scale of Wasserstein distances will depend on the distributions being compared. However, the important insights from Table 2-3 and Figure 3 are the trends.

| Problem | Core | Representation Size / Number of samples | Wasserstein Distance (mean ± std. dev.) | Run time (ms) (mean ± std. dev.) |
|---|---|---|---|---|
| Monte Carlo Convergence Challenge | Laplace | 16 | $0.00457 \pm 0.00004$ | $0.011 \pm 0.001$ |
| **Monte Carlo Convergence Challenge** | **Laplace** | **32** | $\mathbf{0.00167 \pm 0.00007}$ | $\mathbf{0.020 \pm 0.004}$ |
| Monte Carlo Convergence Challenge | Laplace | 64 | $0.00097 \pm 0.00008$ | $0.034 \pm 0.004$ |
| Monte Carlo Convergence Challenge | Laplace | 128 | $0.00065 \pm 0.00018$ | $0.044 \pm 0.003$ |
| Monte Carlo Convergence Challenge | Laplace | 256 | $0.00054 \pm 0.00021$ | $0.073 \pm 0.002$ |
| Monte Carlo Convergence Challenge | Laplace | 2048 | $0.00042 \pm 0.00020$ | $0.531 \pm 0.008$ |
| Monte Carlo Convergence Challenge | Traditional Monte Carlo | 4 | $0.13781 \pm 0.06187$ | $0.077 \pm 0.049$ |
| Monte Carlo Convergence Challenge | Traditional Monte Carlo | 256 | $0.02136 \pm 0.01130$ | $0.086 \pm 0.011$ |
| Monte Carlo Convergence Challenge | Traditional Monte Carlo | 1152 | $0.00844 \pm 0.00363$ | $0.155 \pm 0.035$ |
| Monte Carlo Convergence Challenge | Traditional Monte Carlo | 2048 | $0.00659 \pm 0.00277$ | $0.243 \pm 0.109$ |
| Monte Carlo Convergence Challenge | Traditional Monte Carlo | 4096 | $0.00437 \pm 0.00205$ | $0.381 \pm 0.125$ |
| Monte Carlo Convergence Challenge | Traditional Monte Carlo | 8192 | $0.00307 \pm 0.00123$ | $0.727 \pm 0.199$ |
| Monte Carlo Convergence Challenge | Traditional Monte Carlo | 16000 | $0.00270 \pm 0.00139$ | $1.260 \pm 0.234$ |
| **Monte Carlo Convergence Challenge** | **Traditional Monte Carlo** | **32000** | $\mathbf{0.00158 \pm 0.00068}$ | $\mathbf{2.277 \pm 0.346}$ |
| Monte Carlo Convergence Challenge | Traditional Monte Carlo | 128000 | $0.00086 \pm 0.00035$ | $8.225 \pm 0.849$ |
| Monte Carlo Convergence Challenge | Traditional Monte Carlo | 256000 | $0.00067 \pm 0.00038$ | $14.645 \pm 1.330$ |

Table 2: Results show the mean Wasserstein distance, the run time and the factor increase in dynamic instructions required, with their 1-standard deviation errors for the Monte Carlo Convergence Challenge example. We have highlighted in bold the best overall configuration for Laplace and the close-to-equivalent results Monte Carlo configurations. Traditional Monte Carlo takes approximately $113.85\times$ more time than Laplace. To have better accuracy than the Laplace result up to 1-standard deviation and 2-standard deviations, Traditional Monte Carlo requires approximately 128,000 samples and 256,000 samples respectively. These take $411.25\times$ and $732.35\times$ more time than Laplace, respectively.

| Problem | Core | Representation Size / Number of samples | Wasserstein Distance (mean ± std. dev.) | Run time (ms) (mean ± std. dev.) |
|---|---|---|---|---|
| Poiseuille's Law for Blood Transfusion | Laplace | 16 | $0.00085 \pm 0.00001$ | $0.051 \pm 0.003$ |
| **Poiseuille's Law for Blood Transfusion** | **Laplace** | **32** | $\mathbf{0.00033 \pm 0.00003}$ | $\mathbf{0.173 \pm 0.006}$ |
| Poiseuille's Law for Blood Transfusion | Laplace | 64 | $0.00017 \pm 0.00003$ | $0.412 \pm 0.006$ |
| Poiseuille's Law for Blood Transfusion | Laplace | 128 | $0.00015 \pm 0.00004$ | $1.406 \pm 0.017$ |
| Poiseuille's Law for Blood Transfusion | Laplace | 256 | $0.00015 \pm 0.00004$ | $5.800 \pm 0.055$ |
| Poiseuille's Law for Blood Transfusion | Laplace | 2048 | $0.00014 \pm 0.00005$ | $480.637 \pm 2.663$ |
| Poiseuille's Law for Blood Transfusion | Traditional Monte Carlo | 4 | $0.05379 \pm 0.01818$ | $0.066 \pm 0.019$ |
| Poiseuille's Law for Blood Transfusion | Traditional Monte Carlo | 256 | $0.00699 \pm 0.00245$ | $0.089 \pm 0.031$ |
| Poiseuille's Law for Blood Transfusion | Traditional Monte Carlo | 1152 | $0.00313 \pm 0.00112$ | $0.212 \pm 0.299$ |
| Poiseuille's Law for Blood Transfusion | Traditional Monte Carlo | 4096 | $0.00183 \pm 0.00052$ | $0.345 \pm 0.049$ |
| Poiseuille's Law for Blood Transfusion | Traditional Monte Carlo | 8192 | $0.00118 \pm 0.00038$ | $0.676 \pm 0.204$ |
| Poiseuille's Law for Blood Transfusion | Traditional Monte Carlo | 32000 | $0.00067 \pm 0.00025$ | $2.744 \pm 0.935$ |
| **Poiseuille's Law for Blood Transfusion** | **Traditional Monte Carlo** | **128000** | $\mathbf{0.00033 \pm 0.00009}$ | $\mathbf{8.914 \pm 1.566}$ |
| Poiseuille's Law for Blood Transfusion | Traditional Monte Carlo | 256000 | $0.00023 \pm 0.00008$ | $15.303 \pm 2.611$ |
| Poiseuille's Law for Blood Transfusion | Traditional Monte Carlo | 512000 | $0.00017 \pm 0.00004$ | $27.690 \pm 3.361$ |
| Poiseuille's Law for Blood Transfusion | Traditional Monte Carlo | 640000 | $0.00015 \pm 0.00005$ | $33.853 \pm 1.270$ |

Table 3: Results show the mean Wasserstein distance, the run time and the factor increase in dynamic instructions required, with their 1-standard deviation errors for the Poiseuille's Law for Blood Transfusion example. We have highlighted in bold the best overall configuration for Laplace and the close-to-equivalent results Monte Carlo configurations. Traditional Monte Carlo takes approxiamately $51.53\times$ more time than Laplace with $r = 32$. To have better accuracy than the Laplace result up to 1-standard deviation and 2-standard deviations, Traditional Monte Carlo requires approximately 512,000 samples. This takes $160.06\times$ more time than Laplace.

## F.2 Comparing Histograms of Output Distributions

Figure 4 shows histograms of output distributions for each of the applications. A key observation from these is that the outcome of Laplace, even with high representation sizes is slightly different from the ground truth distributions. We can note that the distribution produced by Laplace puts higher probability density at the mode, and less on the tails. This is contrasted by the histograms of the Monte Carlo method, where the output distribution eventually approaches the ground truth, as the number of samples is increased.

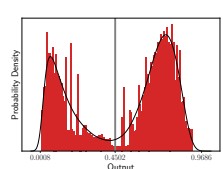
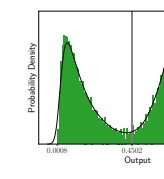
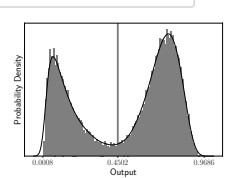
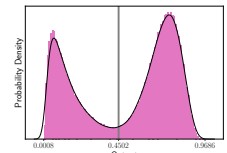

Deterministic solution — Ground truth

(a) Monte Carlo Convergence Challenge application. Laplace. Representation size: 32, Wasserstein Distance: 0.00166, Run time: 19 μs.

(b) Monte Carlo Convergence Challenge application. Laplace. Representation size: 256, Wasserstein Distance: 0.00068, Run time: 75 μs.

(c) Monte Carlo Convergence Challenge application. Traditional Monte Carlo simulation. Number of samples: 32,000, Wasserstein Distance: 0.00119, Run time: 2.07 ms.

(d) Monte Carlo Convergence Challenge application. Traditional Monte Carlo simulation. Number of samples: 256,000, Wasserstein Distance: 0.00045, Run time: 15.61 ms.

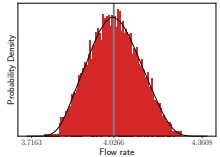
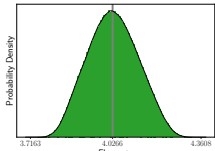
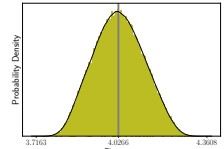
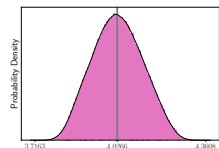

(e) Poiseuille's Law for Blood Transfusion. Laplace. Representation size: 32, Wasserstein Distance: 0.000358, Run time: 171 μs.

(f) Poiseuille's Law for Blood Transfusion. Laplace. Representation size: 256, Wasserstein Distance: 0.000086, Run time: 5.76 ms.

(g) Poiseuille's Law for Blood Transfusion. Traditional Monte Carlo simulation. Number of samples: 128,000, Wasserstein Distance: 0.000207, Run time: 8.66ms.

(h) Poiseuille's Law for Blood Transfusion. Traditional Monte Carlo simulation. Number of samples: 640,000, Wasserstein Distance: 0.000111, Run time: 30.72 ms.

Figure 4: Histograms from example experiments on all applications, showing outputs of Laplace ((a), (b), (e), and (f)) and the Monte Carlo method ((c), (d), (g), and (h)). For the Laplace plots, we have taken 1,000,000 samples from Laplace's internal distribution representation. We set the number of histogram bins to 100 for all cases. The black outline shows a kernel density estimation of the ground truth obtained by Monte Carlo simulations with 1,000,000 samples. The gray vertical lines show the deterministic evaluation, where all uncertain input values and parameters are assumed to have taken their mean value. We also show the minimum and maximum sample values that were for the ground truth.

