# OpenReview forum: "The Monte Carlo Method and New Device and Architectural Techniques for Accelerating It"
_NeurIPS.cc/2024/Workshop/MLNCP — Submitted to MLNCP_

### Official Review · Reviewer_JPnZ · 2024-10-04
**Review of paper number 26**

**Rating:** 5
**Confidence:** 4

**Review:**

This paper provides a brief summary on Monte Carlo method and its recent advances on physics-based non-uniform random number generation. In addition, it also introduces Laplace, a processor microarchitecture that can directly process distribution representations, which has the potential to avoid the use of Monte Carlo simulation.
The paper is written clearly. Here is a list of pros and cons:

**Pros:**
1. This paper identifies the application challenges of Monte Carlo method in practice.
2. The results provides a clear comparison between Monte Carlo simulation and Laplace. They demonstrates Laplace can reach the similar accuracy of Monte Carlo simulation with much shorter run time.

**Cons:**
1. The focus of this paper seems to be on Laplace instead of Monte Carlo method. From the title of this paper, I would expect the paper is accelerating Monte Carlo method with some new hardware techniques.

2. In the experiments, only the classical Monte Carlo method is evaluated. It will be better to include some results from Spot and Grappa for a more complete comparison. Also, there is a large collection of existing works on hardware accelerated Monte Carlo simulation, which hasn't been included in this paper.

3. The implementation setup details of the experiment results needs more clarification. This paper's appendix mentions that it uses Apple M1 Pro as the processor to run Mont Carlo method without parallel execution.
    * Which type of M1-Pro's core is used? Efficient core or Performance core?
    * Is M1-Pro run with battery mode or power plug-in mode?

4. The direct use of rand from Standard C Library seems to be naive pseudo-random number generation. This can easily become the performance bottleneck. The evaluation step of Monte Carlo method is expected to be the most time-consuming step. It will be better to specify the sampling step of this paper's implementation is currently not the bottleneck. It will be helpful to have a time cost breakdown of each step in Monte Carlo mentod.

5. It is not clear if Laplace is a software implementation executed on Apple M1 Pro as well. In Appendix C.3, the paper mentioned Laplace cores that are nowhere described or explained. This makes the comparison unclear.

6. In Section 6, the paper states that much better accuracy needs to be achieved with Monte Carlo method. This is in conflict with what is claimed earlier in the paper. It is suggested to specify the limitation of Laplace as well and more clearly state when Monte Carlo method will be better than Laplace.

7. In Figure 1, Laplace shows larger variance when increasing the representation size. In this paper, authors claim that it is due to sampling variance. More details are needed to provide a clearer root cause explanation.

---

### Official Review · Reviewer_yaww · 2024-10-04
**A consideration of the efficiency of computing the Monte Carlo method**

**Rating:** 5
**Confidence:** 3

**Review:**

The author's submission seeks to examine different means of computing the Monte Carlo method, culminating in a comparison of a traditional method versus a Laplace microarchitecture approach. They provide an extensive supplementary materials section establishing the mathematical basis of Monte Carlo as well as their experiment.

The author's work is high quality and well appreciated. However, overall, the submission has key shortcomings resulting in my review rating of Marginally below acceptance threshold.
- The author's contribution of a comparison between traditional and Laplace architecture Monte Carlo methods was unclear until page four at the methods section. While the description of Monte Carlo methods at large is high quality, the paper is vague as to what it is contributing for the majority of the main body text.
- While PPRVGs may offer a future direction for Monte Carlo, and perhaps an alignment with MLNCP, why are they included in this submission? They do not seem to be used in the analysis and provide a disconnect as to where the author's are going in their narrative.
- The relevance and alignment with MLNCP is non-obvious. The submission itself offers no description as to the connection to the machine learning field. And so is the intended impact to relate to sampling data distributions, probabilistic learning, stochastic neural networks, something else?

Overall, the work can provide contributions to the ML community, but it is recommended the narrative be honed before being publication ready.

---

### Decision · Program_Chairs · 2024-10-10

Reject